# Learning to Group Auxiliary Datasets for Molecule

Tinglin Huang[1]     Ziniu Hu[2]     Rex Ying[1]

[1]Yale University,  [2]University of California, Los Angeles

## Abstract

The limited availability of annotations in small molecule datasets presents a challenge to machine learning models. To address this, one common strategy is to collaborate with additional auxiliary datasets. However, having more data does not always guarantee improvements. Negative transfer can occur when the knowledge in the target dataset differs or contradicts that of the auxiliary molecule datasets. In light of this, identifying the auxiliary molecule datasets that can benefit the target dataset when jointly trained remains a critical and unresolved problem. Through an empirical analysis, we observe that combining graph structure similarity and task similarity can serve as a more reliable indicator for identifying high-affinity auxiliary datasets. Motivated by this insight, we propose ***MolGroup***[1], which separates the dataset affinity into task and structure affinity to predict the potential benefits of each auxiliary molecule dataset. MolGroup achieves this by utilizing a *routing mechanism* optimized through a *bi-level optimization framework*. Empowered by the meta gradient, the routing mechanism is optimized toward maximizing the target dataset's performance and quantifies the affinity as the gating score. As a result, MolGroup is capable of predicting the optimal combination of auxiliary datasets for each target dataset. Our extensive experiments demonstrate the efficiency and effectiveness of MolGroup, showing an average improvement of 4.41%/3.47% for GIN/Graphormer trained with the group of molecule datasets selected by MolGroup on 11 target molecule datasets.

## 1 Introduction

Predicting and understanding molecular properties is a fundamental task in biomedical and chemical fields [51, 14, 23, 55], such as evaluating the toxicity of new clinical drugs, characterizing the binding results for the inhibitors of human $\beta$-secretase, and predicting the thermodynamic property of organic molecules. In recent years, machine learning-based methods have shown promise in this domain [19, 57, 30, 63, 50]. However, labeling molecules requires expensive real-world clinical trials and expert knowledge, making it difficult to collect a large and diverse labeled dataset for training. This limited availability of labeled data poses a challenge for traditional machine learning methods, which struggle to effectively generalize on small

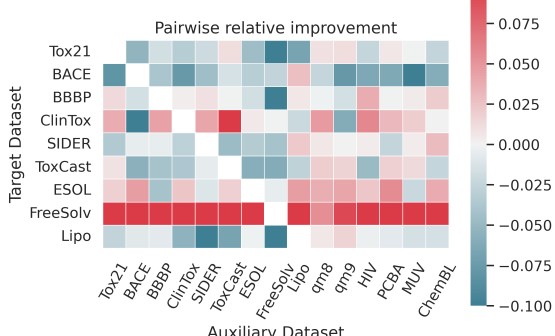

Figure 1: Relative improvement of using the combination of target dataset and auxiliary dataset over only using target dataset: $(\text{perf}(a, b) - \text{perf}(a))/\text{perf}(a)$, where $a$ is target dataset and $b$ is auxiliary dataset.

---

[1]Source code is available at https://github.com/Graph-and-Geometric-Learning/MolGroup.

37th Conference on Neural Information Processing Systems (NeurIPS 2023).

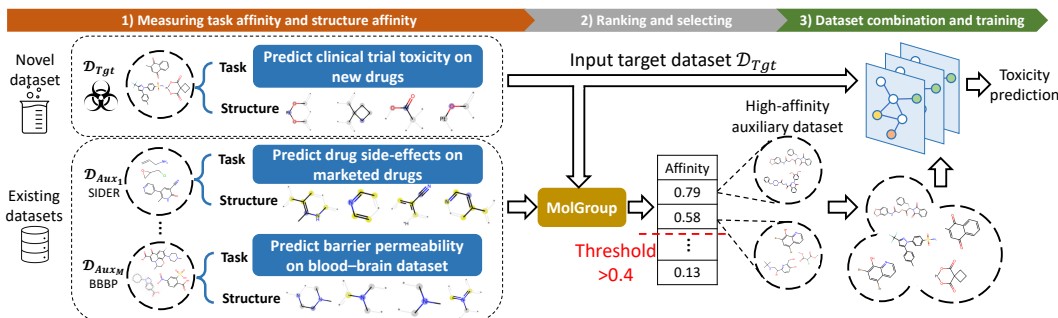

Figure 2: Overview of our proposed MolGroup to identify the auxiliary datasets with high affinity. (1) Measure the affinity between the given new molecule dataset and the existing molecule datasets in terms of task and structure. (2) Rank the auxiliary datasets and filter them according to a predetermined threshold. (3) Train the model with the combination of the selected auxiliary datasets and the target dataset.

datasets. To alleviate this, many methods resort to incorporating additional sources of data to the learning process [19, 6, 29, 41, 57], such as augmenting the current drug data with other known drugs of toxicity. This approach can significantly improve performance by introducing useful out-of-distribution knowledge [58, 54] and emphasizing attention on relevant features [12, 38].

Unfortunately, negative transfer can occur when the included data competes for model capacity or when the model fails to learn a shared representation [60, 45, 20]. This phenomenon is also evident in our empirical study, as illustrated in Figure 1. We pair each target dataset with every other dataset and measure the relative improvement achieved by the combination[2]. The results demonstrate how auxiliary datasets positively or negatively affect the target dataset. For instance, ToxCast greatly improves the performance of ClinTox, while FreeSolv consistently degrades all other datasets' performance. These findings highlight the presence of underlying affinity between different datasets, where auxiliary datasets with high affinity yield greater benefits[3]. In light of this, *how can we measure the affinity of auxiliary datasets to a target dataset?*

**Measured by both structure and task similarity** — a clear answer is well supported by our preliminary analysis (Section 3). We compare the similarity between molecule datasets based on task and graph structure separately and examine their correlation with the relative improvement. Our results confirm that combining both task and structure similarity leads to a stronger correlation with relative improvement. However, existing works on task grouping [12, 1, 60, 45] primarily model task similarities. They rely on exhaustively searching or examining the effect of one task on another. Without further incorporating structure affinity, these methods fail to achieve better performance.

**Present work.** In this paper, we focus on designing a dataset grouping method for molecules. Here we propose MolGroup, a routing-based molecule grouping method. As shown in Fig. 2, MolGroup involves calculating the affinity scores of each auxiliary dataset based on the graph structure and task information, and selecting the auxiliary datasets with high affinity. The selected datasets are then combined and fed into the downstream model. The key idea is to adopt a *routing mechanism* in the network and learn to route between auxiliary-specific or target-specific parameters through a *bi-level optimization framework*. The routing mechanism is optimized toward maximizing the target dataset's performance, and its learned gating score serves as the affinity score. This score is calculated by measuring and combining the task and structure affinity between datasets. Our main contributions can be summarized below:

- We provide an empirical study to investigate how different molecule datasets affect each other's learning, considering both task and structure aspects.
- We propose MolGroup, a molecule dataset grouping method, which outperforms other methods on 11 molecule datasets and achieves an average improvement of 4.41%/3.47% for GIN/Graphormer.

---

[2]More experimental details can be found in Appendix C.

[3]In this paper, "affinity" refers to the impact of one dataset on the performance of another dataset. High-affinity datasets benefit the target dataset, while low-affinity datasets negatively affect the target dataset. It is distinct from "similarity" in this context.

## 2 Related Work

**Molecule representation pretraining.**  Recently, the pretrain-finetune paradigm has been extensively applied to learn the representation of molecules [57, 19, 16, 37, 52, 55]. These methods involve pretraining a graph neural network on a large-scale molecule dataset through contrastive [35, 19, 47, 49, 56, 49] or generative [16, 19, 63] proxy tasks, and finetuning the model on specific downstream task. In our experiments, we demonstrate that incorporating high-affinity auxiliary datasets can significantly improve the performance of pretrained models on downstream target datasets. In addition, our proposed MolGroup can seamlessly integrate with the pretrained models, and the grouping results obtained by the lightweight surrogate model can effectively generalize.

**Task grouping.**  Many applications use multi-task learning [29, 8, 59, 28, 41, 20] to reduce the inference time required to conduct multiple tasks individually or enhance the performance of the tasks with limited labels. To avoid negative transfer, previous methods have been proposed to separate tasks with high affinity. Early studies [60, 45, 22] formulate this as an integer program problem and solve it through optimization. Recent works [12, 44] learn the underlying affinity between different tasks by quantifying the effect of the task's gradient or active learning strategy. In this paper, we focus on grouping the auxiliary molecule datasets that can maximize the performance of the target dataset. By incorporating both structure and task information, our method can achieve better performance.

While MolGroup and MTG-Net [44] are both related to meta-learning approaches for solving grouping problems, the specific objectives differ. MTG-Net formulates the grouping problem as a meta-learning problem and learns the relationship between tasks with a few meta-training instances, while our approach uses the meta gradient to optimize the routing mechanism instead of applying the meta-learning paradigm.

**Conditional computation.**  A plethora of studies applies conditional computation for scaling up model capacity or adapting computation to input [4, 39, 15, 26, 42, 62, 3]. For instance, Mixture-of-Experts layer [42, 26, 36] introduces the routing mechanism to route each token of a sequence to a subset of experts, which increases the number of parameters without incurring too much computational cost. Some methods apply conditional computation to limit the attention operation for speeding up the inference [10, 3, 62, 46]. Another line of work in multilingual modeling [61] includes the routing mechanism in each layer to enforce the model to learn language-specific behavior, which can be considered as a simplified neural architecture search framework. Different from these existing works, the routing mechanism in MolGroup aims to endow the model with the ability to determine the fusion significance of two datasets' parameters, which is used as the affinity score.

## 3 Understanding Relationship between Molecule Datasets

Molecule datasets comprise two critical aspects of information: the structural characteristics of the molecules and the associated predictive tasks. Using this insight, we analyze the relationship between molecule datasets by dividing them into these two dimensions. We conduct an empirical study that links the discrepancies in structure and task between datasets to changes in performance when they train together. Molecule structure and predictive task are quantified as the distribution of fingerprint [5, 7] and task embedding [1]:

- **Fingerprint distribution**: Fingerprint is a multi-hop vector used to character a molecule's structure, with each bit representing a certain chemical substructure. We first convert all the instances into MACC [11] fingerprint, then obtain the fingerprint distribution of the dataset by computing the occurrence frequency of each bit.
- **Task embedding**: We apply Task2vec [1] to quantify task representation which uses Fisher Information Matrix associated with a pretrained probe network. GIN [53] is applied as the probe network with extracted atom and bond features as input [18], and the Monte Carlo algorithm is employed to approximate the Fisher Information Matrix, following the suggested implementation.

Considering the asymmetric nature of the impact between molecule datasets, we use asymmetric KL divergence as the similarity metric [9, 25]. Structural similarity between datasets is measured by fingerprint distribution similarity, while task similarity is measured by task embedding similarity. We plot the regression curve and calculate the Pearson correlation between relative improvement and

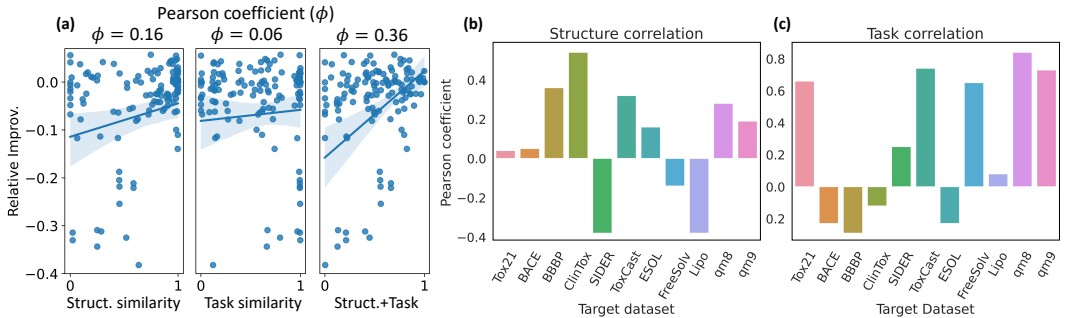

Figure 3: **(a)** Regression curves between relative improvement and the measures of structure similarity, task similarity, and their mixing. **(b,c)** Pearson correlation between relative improvement and similarity of fingerprint distribution/task embedding of each molecule dataset individually.

structural/task similarity over all the combination pairs in Fig.3(a). Additionally, we compute the structural and task similarity between each target dataset and the other 14 datasets individually. We then calculate the Pearson correlation between the similarity scores and the corresponding relative improvement for each target dataset, as shown in Fig.3(b) and (c).

**Combination of task and structure leads to better performance.** We combine the structure similarity and task similarity for each dataset pair by adding them together, and calculate the Pearson coefficient between the combined similarity and relative improvement in Fig.3(a). This combined similarity demonstrates a stronger correlation with the performance compared to using the two similarities separately, demonstrating the effectiveness of incorporating both structure and task information. Despite this, the existing methods of multi-task grouping primarily focus on leveraging the underlying interdependence among tasks, without explicitly incorporating the structure feature.

**Structure and task are compensatory.** Based on Fig.3(b) and (c), we can observe that the structure and task correlation exhibit a compensatory relationship. For instance, a dataset may present a low structural correlation but a high correlation in task similarity, as observed in Tox21, where the structural correlation is 0.04, while the task correlation is 0.66. This suggests that these two sources of information contribute to the performance gain in a complementary manner, such that the inadequacy of one source does not preclude performance improvement facilitated by the other. This observation also demonstrates that the affinity of an auxiliary dataset to a target dataset should be determined by the discrepancies in both structure and task.

**Both similar and dissimilar structures and tasks can benefit target dataset.** According to Fig.3(b) and (c), 8 out of 11 cases in the structural analysis and 7 out of 11 cases in the task analysis show a positive correlation, indicating that the auxiliary datasets with similar structures and predicted tasks can effectively augment the target dataset and improve its performance.

There are also negatively related cases, such as SIDER in Fig.3(b) and BACE in Fig.3(c), indicating that larger discrepancies in graph structure and task can potentially lead to greater improvement. This observation aligns with prior studies on out-of-distribution generalization [58, 54] which demonstrate that dissimilar graph structures and tasks can improve the performance in small datasets with limited annotations. These findings confirm that the additional information required from the other sources of data varies across different target datasets, and both similar and dissimilar structures and tasks from the auxiliary datasets can potentially benefit the target dataset.

Based on the above observations, we argue that an ideal criterion for selecting auxiliary datasets should incorporate measures of both structure and task similarity to the target dataset, and balance these two features. In this study, we propose MolGroup, a routing-based grouping method that allows us to capture the task and graph structure affinity between two molecule datasets.

## 4 Grouping Molecule Datasets with MolGroup

In this section, we introduce MolGroup, a molecule dataset grouping method. MolGroup includes two modules: (1) **routing mechanism** used to quantify the affinity between two datasets, and (2)

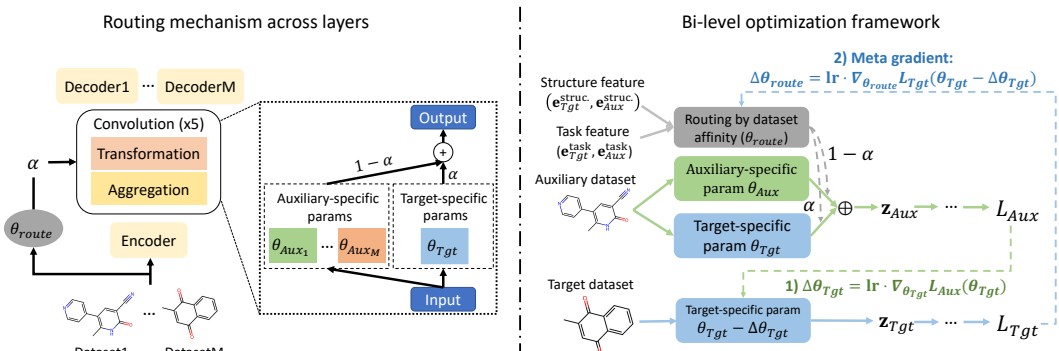

Figure 4: Overview of routing mechanism and bi-level optimization framework.

**bi-level optimization framework** used to update the routing mechanism through meta gradient. Unlike previous methods, the proposed routing function can comprehensively measure the affinity from two perspectives: task and graph structure. We formally define these two modules in Section 4.1 and Section 4.2, and explain our selection process in Section 4.3. More details can be found in Appendix B.

## 4.1 Routing Mechanism

The objective of MolGroup is to determine the affinity between a target dataset $\mathcal{D}_T$ and a set of auxiliary datasets $\{\mathcal{D}_{Aux}\}_M$. To this end, we propose to use the routing mechanism that dynamically allocates the impact of each auxiliary dataset on the target dataset across the network's sub-layers. Intuitively, this routing mechanism allows us to control the contribution of the auxiliary datasets to the final predictions, ensuring the incorporation of the high-affinity auxiliary datasets while mitigating any interference. We first formulate the graph convolution process of a GNN at $l$-th layer as:

$$\mathbf{z}^{(l+1)} = f_\theta^{(l)}(\mathbf{z}^{(l)}), \tag{1}$$

where $\mathbf{z}^{(l+1)}$ is the representation of the input batch at $l$-th layer, and $f_\theta^{(l)}(\cdot)$ denotes the convolution function with parameter $\theta$ at $l$-th layer. As illustrated in Fig. 4, in MolGroup, every dataset is assigned a specific parameter $\theta_T, \theta_1, \cdots, \theta_M$, and we apply the routing mechanism to the auxiliary dataset's convolution process while keeping the convolution process of the target dataset unchanged. Specifically, given an input batch of the target dataset $\mathcal{B}_T$ and $m$-th auxiliary dataset $\mathcal{B}_m$, the convolution with routing mechanism for $m$-th auxiliary dataset at $l$-th layer is calculated as:

$$\mathbf{z}_m^{(l+1)} = \alpha_m f_{\theta_T}^{(l)}(\mathbf{z}_m^{(l)}) + (1 - \alpha_m)f_{\theta_m}^{(l)}(\mathbf{z}_m^{(l)}), \tag{2}$$

$$\text{with} \quad \alpha_m = g_m(\mathcal{B}_T, \mathcal{B}_m), \tag{3}$$

where $\mathbf{z}_m^{(l)}$ is the representation of $m$-th auxiliary dataset at $l$-th layer, and $\alpha_m$ is the gating score generated by the routing function $g_m(\cdot)$. The routing function endows the model with the capability of determining the impact of the auxiliary dataset on the target dataset's parameter. A closed gate suggests that the backward gradient has a minor impact on $\theta_T$, particularly when $\alpha_m = 0$, indicating no interaction between the auxiliary dataset and the target dataset. Conversely, an open gate encourages the gradient from the auxiliary dataset to operate on $\theta_T$, particularly when $\alpha_m = 1$, indicating a hard-parameter sharing architecture.

The calculation of $g(\cdot)$ is parameterized using two modules to capture task affinity and structure affinity. For task affinity, we assign learnable embeddings $\mathbf{e}_T^{\text{task}}$ and $\mathbf{e}_m^{\text{task}}$ for the target dataset and auxiliary dataset respectively, which updated through optimization. For structure affinity, we first extract the fingerprint feature $x^{\text{fp}}$ for each molecule instance in the input batch. Then, we embed it using a weight matrix and apply Set2Set [48] to obtain a single embedding $\mathbf{e}^{\text{struct.}}$:

$$\mathbf{e}^{\text{struct.}} = \text{Set2Set}\left(\{\mathbf{W}x^{\text{fp}}\}_n\right), \tag{4}$$

where $\mathbf{W}$ is a learnable weight matrix, and $n$ is the number of the instance in the batch. Task affinity score and structure affinity score are computed as the cosine similarity and fused as the dataset affinity

**Algorithm 1:** Iterative filtering process with MolGroup

---

**Input:** Target dataset $\mathcal{D}_T$ with $N$ training instances; Candidate auxiliary datasets $\{\mathcal{D}_A\}_M$; GNN model with specific parameters $\theta_T, \{\theta_m\}_M$ for each dataset and routing mechanism $g(\cdot)$; Number of iterations $R$; Number of epochs $E$; Learning rate $lr$; Batch size $B$.

// Filtering round
**for** $r \leftarrow 1, \cdots, R$ **do**
    Random initialize $\theta_T, \theta_1, \cdots, \theta_M$ and $g(\cdot)$.
    $\phi_1, \cdots, \phi_M \leftarrow 0$. // Affinity scores
    $I \leftarrow NM//B$. // Number of iteration in each epoch
    // Training epoch
    **for** $e \leftarrow 1, \cdots, E$ **do**
        // Training step
        **for** $iter \leftarrow 1, \cdots, I$ **do**
            Sample mini-batch $\{\mathcal{B}_T, \mathcal{B}_1, \cdots, \mathcal{B}_M\}$ from current datasets.
            Obtain losses $l_T, \{l_m\}_M$ and affinity scores $\{\alpha_m\}_M$ by feeding mini-batch to GNN.
            // Bi-level optimization framework
            1) $\theta_T(\{\alpha_m\}_M) \leftarrow \theta_T - lr \cdot \nabla_{\theta_T} \sum_m^M l_m$.
            2) Update $g(\cdot)$ through target dataset's loss function with $\mathcal{B}_T$ and $\theta_T(\{\alpha_m\})_M$.
            Update all the parameters $\theta_T, \theta_1, \cdots, \theta_M$ through $l_T, l_1, \cdots, l_M$.
            **if** $e == E$ **then**
                // Average affinity scores in final epoch
                **for** $m \leftarrow 1, \cdots, M$ **do**
                    $\phi_m \leftarrow \phi_m + \alpha_m/I$.
                **end**
            **end**
        **end**
    **end**
    // Remove datasets according to threshold
    $\{\mathcal{D}_A\}_{M'} \leftarrow \{\mathcal{D}_{A_m}|\phi_m \geq 0.6\}$.
    $M \leftarrow M'$.
**end**
**Output:** Auxiliary datasets with high affinity $\{\mathcal{D}_A\}_M$.

---

score, which can be formulated as:

$$\alpha_m = g_m(\mathcal{B}_T, \mathcal{B}_m) = \sigma\left(\lambda \mathbf{e}_T^{\text{task}} \cdot \mathbf{e}_m^{\text{task}} + (1-\lambda)\mathbf{e}_T^{\text{struct.}} \cdot \mathbf{e}_m^{\text{struct.}}\right), \tag{5}$$

where $\sigma(\cdot)$ is the logistic-sigmoid function, and $\lambda$ is a hyperparameter used to balance these two affinity scores. It can be found that the task affinity score is determined globally during training, while the structure affinity score is computed at a per-step level of granularity. To ensure stable learning, a high value for $\lambda$ is suggested. Further analysis can be found in Appendix E. We average the dataset affinity score over a continuous subset of training steps to obtain a final affinity score.

## 4.2 Bi-level Optimization Framework

The learning of the routing mechanism depends on how the auxiliary dataset's gradient affects the target dataset's performance. However, optimizing this routing mechanism in a target dataset-aware way is challenging since it is only used during the forward pass of the auxiliary dataset. To address this, we propose to use a bi-level [34] optimization framework to incorporate the guidance of the target dataset. The framework utilizes the target dataset's performance, using the parameters updated by the auxiliary dataset, as a signal to guide the learning of the routing mechanism.

Equipped with the routing mechanism, the overall optimization can be formulated as:

$$\min_{\theta_T} L_T(\mathcal{B}_T; \theta_T), \tag{6}$$

$$\min_{\theta_T, \theta_m} L_m(\mathcal{B}_m; \theta_T, \theta_m, \alpha_m), \tag{7}$$

where $L_T$ and $L_m$ denote the loss function of the target dataset and auxiliary dataset. It can be found that the optimization of $\theta_T$ is closely linked to the auxiliary task and its relevance is determined by the routing function. We explicitly represent such dependency as $\theta_T(\alpha_m)$. With the goal of obtaining an optimal gating score $\alpha_m$ parameterized by $g_m(\cdot)$, given $M$ auxiliary datasets, we further formulate the optimization as a bi-level framework:

$$\min_{\alpha_1,\cdots,\alpha_M} L_T(\mathcal{B}_T; \theta_T(\{\alpha_m\}_M)), \tag{8}$$

$$\text{where} \quad \theta_T(\{\alpha_m\}_M) = \arg\min_{\theta_T} \sum_m^M L_m(\mathcal{B}_m; \theta_T, \theta_m, \alpha_m), \tag{9}$$

As illustrated in Fig.4, the update process of the routing function is split into two steps: first, the model parameters except the routing function are updated using the gradient from the auxiliary tasks; second, we reuse this computation graph and calculate the meta gradient of the routing mechanism.

### 4.3 Iterative Filtering and Selection

We adopt an iterative filtering process to perform the selection of auxiliary datasets. Initially, all datasets are trained together in a model equipped with the routing mechanism for several epochs. At the final epoch, we average the generated affinity score across all training steps and take it as the final dataset affinity. Datasets with an affinity score below the predefined threshold are then filtered out. This process is repeated for $t$ iterations, and in the last round of filtering, the top-$k$ remaining auxiliary datasets are selected based on their affinity scores. Finally, these selected datasets are combined with the target dataset and used for training the model. The pseudo-code is presented in Algo.1.

**Complexity.** Given $M$ candidate auxiliary datasets, we combine them with the target dataset and use MolGroup for grouping. At each training step, we sample data from each dataset with equal probability to form a mini-batch, ensuring that each dataset contributes equally to the training process. The total number of training instances is set as $N \times M$ to cover all instances in the target dataset, where $N$ is the target dataset size. Therefore, the training complexity of each round is $O(NM)$. This approach is efficient as the number of auxiliary datasets is typically filtered, and further analysis is provided in Section 5.2.

## 5 Experiments

To show the effectiveness of MolGroup, we apply it to 15 molecule datasets with varying sizes and compare it with 8 baseline models. Moreover, we conduct some further studies including an ablation study and efficiency comparison, and conclude some findings regarding the grouping results. The statistics of the involved datasets are summarized in Appendix A, the experimental setting is detailed in Appendix C, and the extensive experimental results can be found in Appendix D.

### 5.1 Dataset Grouping Evaluation

**Baselines.** We compare MolGroup with three classes of approaches: 1) search-based methods, 2) grouping-based methods, and 3) the methods that train on all the datasets. Regarding the search-based methods, we apply beam search [32, 13] and consider two kinds of criteria, i.e., the target dataset's performance on the validation set (P), and the combination of the performance and the difference of fingerprint distribution (P+S). We select the candidates with the highest criterion value and train the model for a few epochs at each selection step to speed up the search process. As for the grouping-based methods, we apply TAG [12] and Task2vec [1], which calculates the pairwise affinity using the gradient-based strategy. For the last class of methods, we consider Unweighted Averages(UA), Gradnorm [8], and MTDNN [27], where MTDNN selects a subset of datasets through the task discriminator. Furthermore, we present the results of Pretrain-Finetune strategy [19, 57], where the model is first trained on PCQM4Mv2 [17] and then finetuned on the downstream dataset.

**Dataset.** Our study utilizes 15 molecule datasets of varying sizes obtained from MoleculeNet [51, 18] and ChemBL [33], which can be categorized into three groups: medication, quantum mechanics, and chemical analysis. Our focus is solely on the small molecule datasets that have less than 10,000

instances in their training sets, totaling 11 target datasets. We follow the original split setting, where qm8 and qm9 are randomly split, and scaffold splitting is used for the others.

**Architectures.** All the model architectures include a standard encoder-decoder, where GIN [53] is used as the encoder in MolGroup and other baseline methods that explicitly group datasets. We evaluate the selected auxiliary datasets using GIN and the SOTA model Graphormer [57]. To accommodate the different tasks, we use a different decoder for each one. The overall training loss is calculated as the unweighted mean of the losses for all included tasks.

| Method | BBBP($\uparrow$) | ClinTox($\uparrow$) | Tox21($\uparrow$) | BACE($\uparrow$) | FreeSolv($\downarrow$) | qm8($\downarrow$) |
|---|---|---|---|---|---|---|
| Only-target | $66.62_{0.028}$ | $56.45_{0.023}$ | $74.23_{0.005}$ | $75.02_{0.026}$ | $3.842_{1.579}$ | $0.0385_{0.001}$ |
| Beam search(P) | $66.02_{0.015}$ | $57.86_{0.068}$ | $\underline{74.71_{0.004}}$ | $67.34_{0.039}$ | $3.271_{0.479}$ | $0.0553_{0.002}$ |
| Beam search(P+S) | $67.69_{0.034}$ | $57.63_{0.036}$ | $74.36_{0.003}$ | $69.74_{0.056}$ | $3.331_{0.287}$ | $\underline{0.0494_{0.000}}$ |
| TAG | $60.66_{0.014}$ | $\underline{57.98_{0.028}}$ | $70.32_{0.006}$ | $70.02_{0.076}$ | $3.922_{0.748}$ | $0.0637_{0.001}$ |
| Task2vec | $\underline{68.18_{0.011}}$ | $47.30_{0.031}$ | $68.00_{0.005}$ | $\underline{74.71_{0.032}}$ | $3.383_{0.766}$ | $0.0635_{0.001}$ |
| MTDNN | $66.56_{0.021}$ | $52.90_{0.039}$ | $71.87_{0.003}$ | $69.91_{0.026}$ | $3.428_{0.733}$ | $0.0523_{0.002}$ |
| UA | $60.41_{0.008}$ | $51.99_{0.078}$ | $68.16_{0.004}$ | $61.75_{0.018}$ | $4.095_{0.334}$ | $0.0625_{0.001}$ |
| Gradnorm | $61.21_{0.007}$ | $53.08_{0.070}$ | $59.40_{0.041}$ | $64.83_{0.028}$ | $4.356_{0.589}$ | $0.0657_{0.006}$ |
| Pretrain-Finetune | $56.59_{0.026}$ | $56.00_{0.037}$ | $50.64_{0.015}$ | $64.80_{0.052}$ | $4.391_{0.043}$ | $0.0637_{0.001}$ |
| MolGroup | $\mathbf{68.36_{0.016}}$ | $\mathbf{59.77_{0.027}}$ | $\mathbf{75.66_{0.004}}$ | $\mathbf{77.33_{0.015}}$ | $\mathbf{3.116_{0.279}}$ | $\mathbf{0.0385_{0.001}}$ |

Table 1: Performance comparison of GIN on target molecule datasets, with $\uparrow$ indicating higher is better and $\downarrow$ indicating lower is better.

Due to the space limit, here we present the comparison results of 6 datasets in Table 1. Our results show that MolGroup outperforms all the baseline methods and consistently improves the performance of the backbone model, with an average relative improvement of 6.47% across all datasets. We observed that MolGroup assigned low-affinity scores to all candidate auxiliary datasets for qm8 and qm9, resulting in no selection of auxiliary datasets for those two target datasets. We attribute this phenomenon to the significant difference between quantum chemistry and other domains. We also observe that UA, Gradnorm, and Pretrain-Finetune methods perform even worse than the model trained only on the target dataset due to the significant distribution shift among these datasets. In addition, the search-based methods' performance is constrained by the limited exploration space, and task grouping methods neglect the data structure difference, leading to suboptimal performance.

| Method | BBBP($\uparrow$) | ClinTox($\uparrow$) | Tox21($\uparrow$) | BACE($\uparrow$) | FreeSolv($\downarrow$) | qm8($\downarrow$) |
|---|---|---|---|---|---|---|
| Only-target | $66.40_{0.019}$ | $77.59_{0.028}$ | $75.97_{0.009}$ | $78.91_{0.023}$ | $2.004_{0.088}$ | $0.0372_{0.002}$ |
| Beam search(P) | $67.53_{0.010}$ | $76.77_{0.117}$ | $76.61_{0.008}$ | $\underline{81.58_{0.038}}$ | $2.129_{0.215}$ | $\underline{0.0473_{0.001}}$ |
| Beam search(P+S) | $67.15_{0.030}$ | $77.70_{0.076}$ | $76.83_{0.007}$ | $80.68_{0.024}$ | $2.312_{0.191}$ | $0.0458_{0.001}$ |
| TAG | $\underline{69.62_{0.008}}$ | $78.08_{0.036}$ | $76.57_{0.006}$ | $80.10_{0.016}$ | $2.038_{0.178}$ | $0.0537_{0.001}$ |
| Task2vec | $66.93_{0.022}$ | $72.92_{0.049}$ | $75.15_{0.007}$ | $80.84_{0.017}$ | $\underline{1.941_{0.123}}$ | $0.0530_{0.001}$ |
| MTDNN | $66.71_{0.039}$ | $71.97_{0.078}$ | $\underline{76.84_{0.008}}$ | $79.79_{0.001}$ | $2.173_{0.321}$ | $0.0512_{0.001}$ |
| UW | $65.37_{0.000}$ | $\underline{80.52_{0.027}}$ | $72.16_{0.009}$ | $75.64_{0.000}$ | $2.405_{0.415}$ | $0.0569_{0.000}$ |
| Gradnorm | $60.40_{0.045}$ | $43.46_{0.083}$ | $55.06_{0.002}$ | $48.00_{0.005}$ | $2.552_{0.595}$ | $0.1694_{0.014}$ |
| MolGroup | $\mathbf{69.66_{0.009}}$ | $\mathbf{81.21_{0.040}}$ | $\mathbf{77.34_{0.006}}$ | $\mathbf{82.94_{0.017}}$ | $\mathbf{1.879_{0.032}}$ | $\mathbf{0.0372_{0.002}}$ |

Table 2: Performance comparison using Graphormer on target molecule datasets.

Additionally, we apply the resulting groups to evaluate the impact of dataset combination using Graphormer as the encoder [57]. We first pretrain Graphormer on PCQM4Mv2 [17], which is the largest graph-level prediction dataset, and then finetune it on the specific downstream datasets with the selected auxiliary datasets. As shown in Table 2, the extensive parameter space and pretraining offer a significant improvement to Graphormer, consistent with the previous studies [57, 43]. Moreover, the dataset combinations found by MolGroup outperform all the baseline methods and provide an average relative improvement of 3.35% across all the target datasets, demonstrating the generalizability of the auxiliary dataset groupings exploited by the lightweight surrogate model GIN.

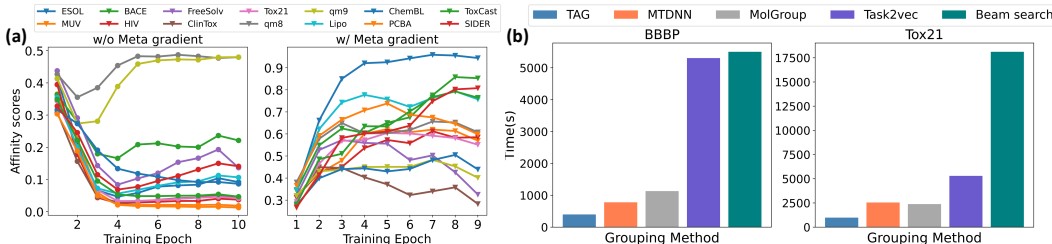

Figure 5: **(a)** Comparison between MolGroup with and without meta gradient on BBBP. **(b)** Efficiency comparison between MolGroup and the other grouping methods.

## 5.2 Further Analysis

**Overall selected results.** In Fig. 6, we visualize the auxiliary datasets with top-3 affinity scores to the target dataset measured by MolGroup. PCBA is selected by most of the datasets due to their diverse structure or useful out-of-distribution information. This phenomenon is supported by Fig. 1, where PCBA demonstrates significant benefits across datasets. We also observe that Tox21 benefits ClinTox and ToxCast, particularly for toxicity-related tasks. Additionally, despite belonging to distinct domains, some datasets exhibit a great affinity to the target dataset, such as qm8 for BBBP and ESOL for Lipo. These findings demonstrate that MolGroup can effectively reveal the underlying affinity between molecule datasets.

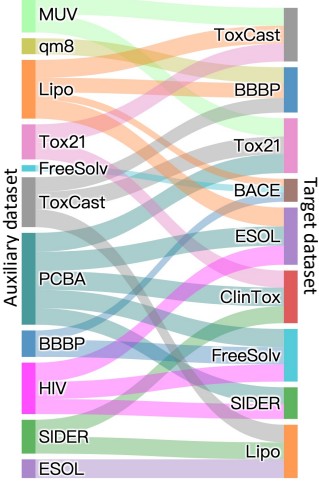

Figure 6: Selected grouping for each target dataset.

**Ablation study.** We conduct an ablation study to verify the impact of the bi-level optimization framework. Specifically, the routing mechanism is solely updated by the auxiliary dataset's task without the guidance of the target dataset. We pick BBBP as the target dataset, and average the gating scores $\alpha_m$ every epoch, which is illustrated in Fig. 5(a). The routing mechanism tends to assign a low weight to the auxiliary-specific parameters (<0.15 in most cases) to increase the dataset-specific capacity, failing to differentiate the affinity of the auxiliary dataset to the target dataset. Conversely, the meta gradient can optimize the routing mechanism toward maximizing the performance of the target dataset, which learns a distinguishable affinity score distribution. Similar phenomena can be found in the other datasets.

**Efficiency Comparison.** We compare the wall-clock time performance of MolGroup with the other grouping methods using the same running environment (see Appendix D). We pick BBBP and Tox21 as the target datasets, and the results are presented in Fig. 5(b). Beam search explores combinations in a greedy manner, but its computational complexity increases exponentially as the search space grows. TAG efficiently computes inter-task affinity by averaging the lookahead loss proposed during training. MTDNN includes all auxiliary datasets and utilizes a task discriminator to select instances, which limits its computational performance based on the number of training instances used. Although we initially input all candidate auxiliary datasets to MolGroup, we only train for a few epochs at each round, and the majority of them are filtered out after the first or second round of training. Specifically, only 6 and 5 out of 14 candidates remain after the first round of filtering for BBBP and Tox21 respectively.

**Comparison with Task2vec: similar task≠benefit.** In Section 3, we conclude that task similarity measured by task embedding is correlated with relative improvement. But Task2vec [1] fails to achieve better performance in our experiments since it doesn't incorporate structure information. It is likely that the datasets with similar tasks can serve as data augmentation and are able to benefit each other during training. We choose Tox21, ToxCast, and ClinTox as examples since they share similar tasks of predicting the absence of toxicity or qualitative toxicity measurement. We combine them pairwise and observe that half of the cases result in negative transfers. To delve deeper into this phenomenon, we average the structure affinity score measured by MolGroup over every epoch and plot the results for positive and negative transfer cases in Fig. 7(a) and (b) respectively.

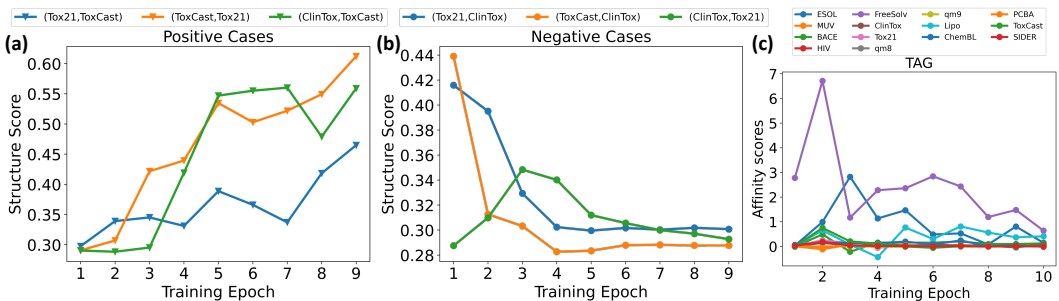

Figure 7: **(a,b)** Structure affinity score of positive and negative cases measured by MolGroup. **(c)** Inter-task affinity score measured by TAG on BBBP.

Our result reveals that the structure affinity consistently converged to a low score during training on negative transfer cases, suggesting a fundamental mismatch between the target task and the auxiliary dataset. The results also demonstrate that the combination of task and structure affinity can lead to a more comprehensive measurement, which is consistent with the insights presented in Sec. 3.

**Comparison with TAG: differential affinity scores perform better.** TAG [12] is the SOTA method in task grouping; however, it fails to select high-affinity auxiliary datasets in our cases. Here we plot the inter-task affinity scores measured by TAG for BBBP in Fig. 7(c). Compared with MolGroup's results shown in Fig. 3(a), we observe that most datasets' affinity scores produced by TAG are homogeneous, centered around 0. Even the dataset with the highest affinity score (FreeSolv) is the negative auxiliary dataset to BBBP, as demonstrated in Fig. 1. One reason is that the lookahead loss of every dataset pair used in TAG can be easily influenced by the other dataset when they are trained together in a shared architecture. In contrast, the specific parameters assigned to each dataset in MolGroup can alleviate this issue and result in more stable training.

**PCBA is an effective booster.** One interesting finding is that dataset PCBA [17] can boost the performance of most of the small molecule datasets, as shown in Table 3. This dataset offers both a diverse range of chemical compounds with unique scaffold structures, comprising over 350,000 training instances, and an extensive collection of 128 bioassay annotations that represent a broad range of biological activities, making it a potent booster for small molecule property prediction tasks that can benefit from both structure and task.

|  | BBBP($\uparrow$) | ClinTox($\uparrow$) | ToxCast($\uparrow$) | Tox21($\uparrow$) | ESOL($\downarrow$) | FreeSolv($\downarrow$) | Lipo($\downarrow$) |
|---|---|---|---|---|---|---|---|
| Only-target | $66.62_{0.028}$ | $56.45_{0.023}$ | $60.69_{0.010}$ | $74.23_{0.005}$ | $1.563_{0.040}$ | $3.842_{1.579}$ | $0.8063_{0.015}$ |
| +PCBA | $67.11_{0.023}$ | $57.77_{0.028}$ | $62.05_{0.007}$ | $74.81_{0.006}$ | $1.463_{0.020}$ | $3.563_{0.989}$ | $0.8021_{0.009}$ |

Table 3: Cases whose performance is improved by PCBA.

# 6   Conclusion

A common strategy to improve the performance of small molecule datasets is to incorporate additional data during training. In this paper, we conduct an empirical study to investigate the relationship between molecule datasets and conclude that an ideal criterion should combine graph structure and task discrepancies. Motivated by this, we propose MolGroup which quantifies the affinity between datasets by a routing mechanism that is optimized by the bi-level optimization framework. We evaluate our method on 15 molecule datasets and compare it with 8 baseline models, and the results demonstrate the effectiveness of MolGroup.

**Limitation.** MolGroup operates at the dataset level and does not consider subdataset-level information. As a result, it cannot fully exploit the potential benefits obtained from specific subdatasets within a large dataset. This limitation could impact the performance of MolGroup on certain datasets, such as qm8 and qm9, where MolGroup fails to group high-affinity auxiliary datasets.

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

# A Dataset Details

Our study utilizes 15 molecule datasets of varying sizes obtained from MoleculeNet [51, 18] and ChemBL [33], which can be categorized into three groups based on their applications or source domains: medication (HIV, SIDER, Lipo, ClinTox), quantum mechanics (qm8, qm9), and chemical analysis (ESOL, FreeSolv, ChemBL, MUV, BACE, BBBP, ToxCast, Tox21, PCBA). As for qm8 and qm9, we randomly sample 3,000 graphs to construct the datasets. We use the original split setting, where qm8 and qm9 are randomly split, and scaffold splitting is used for the others. The small molecule datasets with less than 10,000 instances in the training sets are selected as target molecule datasets, i.e., Tox21, BACE, BBBP, ClinTox, SIDER, ToxCast, ESOL, FreeSolv, Lipo, qm8 and qm9. All the involved datasets can be accessed and downloaded from OGB[4] or MoleculeNet repository[5]. The overall statistics are summarized as follows:

|         | HIV      | PCBA     | Tox21    | BACE     | BBBP     | ClinTox  | MUV      |
|---------|----------|----------|----------|----------|----------|----------|----------|
| #graphs | 41,127   | 437,929  | 7,831    | 1,513    | 2,039    | 1,477    | 93,087   |
| #tasks  | 1        | 128      | 12       | 1        | 1        | 2        | 17       |
| split   | scaffold | scaffold | scaffold | scaffold | scaffold | scaffold | scaffold |
| metric  | ROAUC    | AP       | ROAUC    | ROAUC    | ROAUC    | ROAUC    | AP       |

Table 4: Dataset statistics(1).

|         | SIDER    | ToxCast  | ChemBL   | ESOL     | FreeSolv | Lipo     | qm8    | qm9    |
|---------|----------|----------|----------|----------|----------|----------|--------|--------|
| #graphs | 1,427    | 8,576    | 456,309  | 1,128    | 642      | 4,200    | 3,000  | 3,000  |
| #tasks  | 27       | 617      | 1,310    | 1        | 1        | 1        | 12     | 12     |
| split   | scaffold | scaffold | scaffold | scaffold | scaffold | scaffold | random | random |
| metric  | ROAUC    | ROAUC    | ROAUC    | RMSE     | RMSE     | RMSE     | MAE    | MAE    |

Table 5: Dataset statistics(2).

Here we list a brief description of each dataset's task in this study:

- HIV: Test the ability of compounds to inhibit HIV replication, i.e., estimate whether the molecule is inactive or active.
- PCBA: Predict 128 molecular properties of each compound, where all the properties are cast as binary labels.
- Tox21: Predict the toxicity of each compound on 12 different targets.
- BACE: Predict the binding results (either binding or non-binding) for a collection of inhibitors targeting human $\beta$-secretase 1.
- BBBP: Predict the permeability properties of drugs across the barrier between circulating blood and the brain's extracellular fluid.
- ClinTox: Classify a collection of drug compounds by predicting their (1) clinical trial toxicity (or absence thereof) and (2) FDA approval status.
- MUV: Evaluate 17 molecule properties for around 90 thousand compounds specifically designed to validate virtual screening methodologies.
- SIDER: Predict the drug side effects on 27 system organ classes.
- ToxCast: Predict the toxicity of the compound on over 600 targets.
- ChemBL: Predict the molecular properties of a set of bioactive molecules.
- ESOL: Estimate the water solubility of each molecule.
- FreeSolv: Estimate the hydration-free energy in the water of each molecule.
- Lipo: Evaluate the lipophilicity of the drug molecule.
- qm8/qm9: Estimate the energetic, electronic, and thermodynamic properties of the molecule.

[4] https://ogb.stanford.edu/
[5] https://moleculenet.org/

## B  Implementation Details

**Backbone model settings.** As for GIN [53], we fix the batch size as 128 and train the model for 50 epochs. We use Adam [24] with a learning rate of 0.001 for optimization. The hidden size and number of layers are set as 300 and 5 respectively. We set the dropout rate as 0.5 and apply batchnorm [21] in each layer. All the results are reported after 5 different random seeds.

As for Graphormer [57], we fix the batch size as 128 and train the model for 30 epochs. AdamW [31] with a learning rate of 0.0001 is used as the optimizer. The hidden size, number of layers, and number of attention heads are set as 512, 5, and 8 respectively. We set the dropout rate and attention dropout rate as 0.1 and 0.3. Layernorm [2] is applied across layers. The maximum number of nodes and the distance between nodes in the sampled graph is set as 128 and 5 respectively. The size of position embedding, in-degree embedding, and out-degree embedding are fixed as 512. All the results are reported after 5 different random seeds.

**Grouping method settings.** As for MolGroup, we apply GIN as the encoder. During the training, we randomly select datasets with equal probability and sample data from them to ensure that each dataset contributes an equal number of samples to the constructed mini-batch. We use MACC [11] fingerprint to calculate the structure affinity score. We fix the number of the filtering iterations as 3 and the balance coefficient $\lambda$ is set as 0.9. During each round, the auxiliary datasets with affinity scores below 0.6 will be filtered out. If none of the datasets fall below this threshold, we filter out the dataset with the lowest affinity score. The orthogonal initialization [40] is applied to initialize the learnable task embedding $\mathbf{e}^{\text{task}}$ to make sure that the dot product between the target dataset' embedding and auxiliary dataset's embedding starts with 0. The task embedding size is set as 16, and the number of the processing steps in Set2Set is set as 2. The pseudo-code is presented in Algo.1.

As for beam search, we apply GIN as the encoder. The beam width and search depth are both set to 3. During the search process, we train the model for 3 epochs using each candidate dataset and evaluate its performance using the validation set loss. Additionally, we consider a criterion based on the combination of the difference in fingerprint distribution and performance. Specifically, we average and normalize these two metrics to determine the criterion.

As for Task2vec [1], we employ GIN as the probe network and follow the official implementation[6]. First, we fix the encoder and train the decoder for 10 epochs. Then we apply the Monte Carlo algorithm to compute the Fisher Information Matrix.

As for TAG [12], we use GIN as the encoder and follow the official implementation[7]. TAG involves training all the datasets and computing the lookahead loss between target dataset and auxiliary datasets. The lookahead loss is accumulated over multiple epochs and used as the affinity scores.

As for MTDNN [27], we train all the datasets together for each target dataset and apply an additional task discriminator to classify the source of the dataset for the input instances. We train 50 epochs for GIN and 30 epochs for Graphormer, and the instances in auxiliary datasets that have a probability greater than 0.6 of being classified as the target dataset will be selected.

As for Gradnorm [8], we train all the datasets together and update the weights of the loss based on the gradients of the last shared GNN layer.

## C  Experimental Details

For the preliminary analysis shown in Fig.1, we conduct the study using a set of 15 molecule datasets. Among these datasets, 9 datasets with less than 10,000 instances in the training set are selected as target datasets. We pair the target datasets with every other dataset and measure the relative improvement the combination achieves. To mitigate the issue of varying dataset sizes, we upsample or downsample the training sets of all datasets to ensure an equal number of training instances, specifically 5,000 instances. All the reported results are based on 5 different random seeds.

For the dataset grouping evaluation, we train the model using the combined datasets and assess its performance on the target datasets. We then report the model's performance on the test set using the

---

[6]https://github.com/awslabs/aws-cv-task2vec
[7]https://github.com/google-research/google-research/tree/master/tag

best-performing model selected based on its performance on the validation set. Cross-entropy loss is used for classification tasks and mean squared error loss is used for regression tasks. The overall training loss is calculated as the unweighted mean of the losses for all included tasks. All the reported results are based on 5 different random seeds.

# D   Overall Experimental Results

**Running environment.**   The experiments are conducted on a single Linux server with The Intel Xeon Gold 6240 36-Core Processor, 361G RAM, and 4 NVIDIA A100-40GB. Our method is implemented on PyTorch 1.10.0 and Python 3.9.13.

## D.1   Dataset Grouping Evaluation

Here we present the performance comparison over the other 5 target datasets in Table 6 and Table 7. It can be observed that pretrained Graphormer outperforms GIN significantly, consistent with the previous studies. In addition, MolGroup achieves the best performance in most cases, with an average relative improvement of 3.52% and 3.10% for GIN and Graphormer. However, a notable exception occurs with qm9 where our proposed method is unable to surpass beam search. In this instance, MolGroup assigns low-affinity scores to each auxiliary dataset due to the significant disparity between quantum chemistry and the other domains, as shown in Section D.3. Nonetheless, despite the limited efficiency, beam search is capable of identifying a promising candidate by directly comparing the performance of different groupings. It is worth noting that overall, our proposed MolGroup still achieves better performance compared to the other baseline methods.

Additionally, we attribute the poor performance or even worse results of the Unweighted Average, Gradnorm, and Pretrain-Finetune methods to the significant distribution gap between different datasets. These methods struggle to learn a shared representation that can effectively capture the characteristics of all the datasets.

| Method | SIDER($\uparrow$) | ToxCast($\uparrow$) | ESOL($\downarrow$) | Lipo($\downarrow$) | qm9($\downarrow$) |
|---|---|---|---|---|---|
| Only-target | $59.35_{0.010}$ | $60.69_{0.010}$ | $1.563_{0.040}$ | $0.8063_{0.015}$ | $0.0303_{0.001}$ |
| Beam search(P) | $54.25_{0.024}$ | $63.37_{0.004}$ | $1.431_{0.058}$ | $\underline{0.8073}_{0.011}$ | $0.0456_{0.001}$ |
| Beam search(P+S) | $56.58_{0.038}$ | $61.01_{0.010}$ | $1.476_{0.061}$ | $0.8147_{0.014}$ | $\mathbf{0.0258_{0.000}}$ |
| TAG | $55.64_{0.005}$ | $58.08_{0.003}$ | $\underline{1.417}_{0.066}$ | $0.8170_{0.017}$ | $0.0453_{0.000}$ |
| Task2vec | $55.74_{0.008}$ | $57.45_{0.004}$ | $1.436_{0.050}$ | $0.8078_{0.010}$ | $0.0468_{0.000}$ |
| MTDNN | $55.95_{0.015}$ | $59.02_{0.007}$ | $1.499_{0.027}$ | $0.8155_{0.017}$ | $0.0476_{0.001}$ |
| UA | $\underline{56.26}_{0.009}$ | $59.93_{0.003}$ | $1.480_{0.060}$ | $0.9130_{0.017}$ | $0.0494_{0.000}$ |
| Gradnorm | $55.69_{0.005}$ | $53.31_{0.010}$ | $1.541_{0.091}$ | $1.0755_{0.001}$ | $0.0574_{0.005}$ |
| Pretrain-Finetune | $51.43_{0.009}$ | $51.06_{0.005}$ | $1.480_{0.099}$ | $1.0776_{0.066}$ | $0.0498_{0.010}$ |
| MolGroup | $\mathbf{59.21_{0.013}}$ | $\mathbf{63.91_{0.005}}$ | $\mathbf{1.402_{0.037}}$ | $\mathbf{0.7996_{0.007}}$ | $0.0303_{0.001}$ |

Table 6: Performance comparison of GIN on target molecule datasets, with $\uparrow$ indicating higher is better and $\downarrow$ indicating lower is better.

## D.2   Takeaway

**Grouping more high-affinity datasets improves performance.**   Previous studies on task grouping have assumed that low-order relationships can be an effective indicator of high-order ones. It suggests that combining multiple source datasets, which can benefit the target dataset, leads to better performance when learned together. Our experimental results also confirm this phenomenon. We take eight datasets as examples and add them one by one with the top three datasets having the highest affinity score as measured by MolGroup. Results are presented in Table 8, where Top$\{a, b, \cdots\}$ denotes the combination of auxiliary datasets with top-performing ones. We can find that combining more auxiliary datasets leads to better performance in most cases. Besides, training with high-affinity datasets can significantly reduce the variant of FreeSolv ($1.579 \rightarrow 0.279$), indicating a more robust representation learned from the auxiliary datasets.

| Method | SIDER($\uparrow$) | ToxCast($\uparrow$) | ESOL($\downarrow$) | Lipo($\downarrow$) | qm9($\downarrow$) |
|---|---|---|---|---|---|
| Only-target | $62.05_{0.021}$ | $66.16_{0.004}$ | $1.054_{0.053}$ | $0.7432_{0.032}$ | $0.0273_{0.001}$ |
| Beam search(P) | $60.80_{0.005}$ | $67.36_{0.006}$ | $0.989_{0.058}$ | $0.7640_{0.031}$ | $0.0399_{0.002}$ |
| Beam search(P+S) | $62.67_{0.009}$ | $66.00_{0.003}$ | $1.026_{0.024}$ | $0.7524_{0.017}$ | $\mathbf{0.0262_{0.000}}$ |
| TAG | $63.57_{0.004}$ | $65.40_{0.005}$ | $1.015_{0.045}$ | $0.7507_{0.014}$ | $0.0432_{0.002}$ |
| Task2vec | $60.69_{0.019}$ | $63.28_{0.005}$ | $0.997_{0.028}$ | $0.7562_{0.015}$ | $0.0429_{0.001}$ |
| MTDNN | $61.43_{0.024}$ | $65.05_{0.022}$ | $1.045_{0.003}$ | $0.7716_{0.032}$ | $0.0420_{0.000}$ |
| UW | $58.24_{0.014}$ | $62.71_{0.020}$ | $1.120_{0.084}$ | $0.7887_{0.067}$ | $0.0508_{0.000}$ |
| Gradnorm | $51.65_{0.018}$ | $52.40_{0.007}$ | $1.400_{0.096}$ | $1.0482_{0.029}$ | $0.2154_{0.125}$ |
| MolGroup | $\mathbf{63.75_{0.018}}$ | $\mathbf{68.68_{0.003}}$ | $\mathbf{0.978_{0.047}}$ | $\mathbf{0.7304_{0.018}}$ | $0.0273_{0.001}$ |

Table 7: Performance comparison using Graphormer on target molecule datasets.

| Combinations | ClinTox | Tox21 | FreeSolv | BBBP | BACE | ToxCast | ESOL | Lipo |
|---|---|---|---|---|---|---|---|---|
| Only Target | $56.45_{0.023}$ | $74.23_{0.005}$ | $3.842_{1.579}$ | $66.62_{0.028}$ | $75.02_{0.026}$ | $60.69_{0.010}$ | $1.563_{0.040}$ | $0.8063_{0.015}$ |
| + Top{1} | $57.77_{0.028}$ | $74.81_{0.006}$ | $3.563_{0.989}$ | $67.36_{0.023}$ | $71.27_{0.045}$ | $62.66_{0.002}$ | $1.502_{0.043}$ | $0.8096_{0.014}$ |
| + Top{1,2} | $57.48_{0.032}$ | $75.22_{0.003}$ | $3.462_{0.970}$ | $\mathbf{68.64_{0.012}}$ | $71.13_{0.019}$ | $63.18_{0.006}$ | $1.524_{0.077}$ | $0.8078_{0.011}$ |
| +Top{1,2,3} | $\mathbf{59.77_{0.027}}$ | $\mathbf{75.66_{0.004}}$ | $\mathbf{3.116_{0.279}}$ | $68.36_{0.016}$ | $\mathbf{77.33_{0.015}}$ | $\mathbf{63.91_{0.005}}$ | $\mathbf{1.402_{0.010}}$ | $\mathbf{0.7996_{0.005}}$ |

Table 8: Performance of top-k combinations.

**MolGroup is the hard version of routing mechanism.**   Here we investigate the use of the routing mechanism in the training of the final model. Specifically, we train the encoder with the routing mechanism on all the auxiliary datasets and adjust the influence between datasets instead of iterative filtering, as shown in Table. 9. It can be observed that the model trained without the negative auxiliary datasets (i.e., MolGroup) outperforms significantly the model trained with negative datasets (i.e., Final model with $g(\cdot)$), although when it is equipped with a routing mechanism. Besides, MolGroup can be considered a hard version of the final model including the routing mechanism. Rather than incrementally eliminating negative training signals, it directly filters out negative auxiliary datasets.

| | BBBP($\uparrow$) | ToxCast($\uparrow$) | Tox21($\uparrow$) | ESOL($\uparrow$) | FreeSolv($\downarrow$) |
|---|---|---|---|---|---|
| Only-target | $66.62_{0.028}$ | $60.69_{0.010}$ | $74.23_{0.005}$ | $1.563_{0.040}$ | $3.842_{1.579}$ |
| MolGroup | $68.36_{0.016}$ | $63.91_{0.005}$ | $75.66_{0.004}$ | $1.402_{0.037}$ | $3.116_{0.2790}$ |
| Final Model with $g(\cdot)$ | $66.83_{0.030}$ | $57.06_{0.011}$ | $69.37_{0.018}$ | $3.040_{0.1383}$ | $5.483_{1.3547}$ |

Table 9: Comparison with final model trained on all auxiliary datasets.

## D.3   Case Study

To give an intuition of the learning process of MolGroup, we show the learning curves of the affinity scores in different filtering rounds in Fig. 8. In each round, auxiliary datasets with affinity scores below 0.6 are removed, and if none of the datasets fall below this threshold, the dataset with the lowest affinity score is filtered out. It can be observed that a significant number of auxiliary datasets are removed in the first or second round. Furthermore, the learning curves tend to converge after 6 epochs in most cases. The dataset PCBA remains in the final round in most cases, indicating its general benefit to the other target datasets. We also notice that the majority of auxiliary datasets are assigned high-affinity scores for FreeSolv, as demonstrated in Fig.1, suggesting that all auxiliary datasets contribute positively to its performance.

Additionally, we plot the learning curves for qm8 and qm9, which don't have suggested auxiliary datasets. As shown in Fig. 9, all the auxiliary datasets are assigned affinity scores lower than the threshold for both qm8 and qm9, resulting in the removal of all datasets after the first round. We attribute this to the large discrepancy in the structure and task between quantum chemistry and the other domains such as medication. Molecules in quantum chemistry have diverse structures, including both natural and hypothetical compounds. They are studied to explore the behaviors of various functional groups. On the other hand, molecules in medication are more focused on structure. Their study revolves around predicting and optimizing molecular properties relevant to drug design.

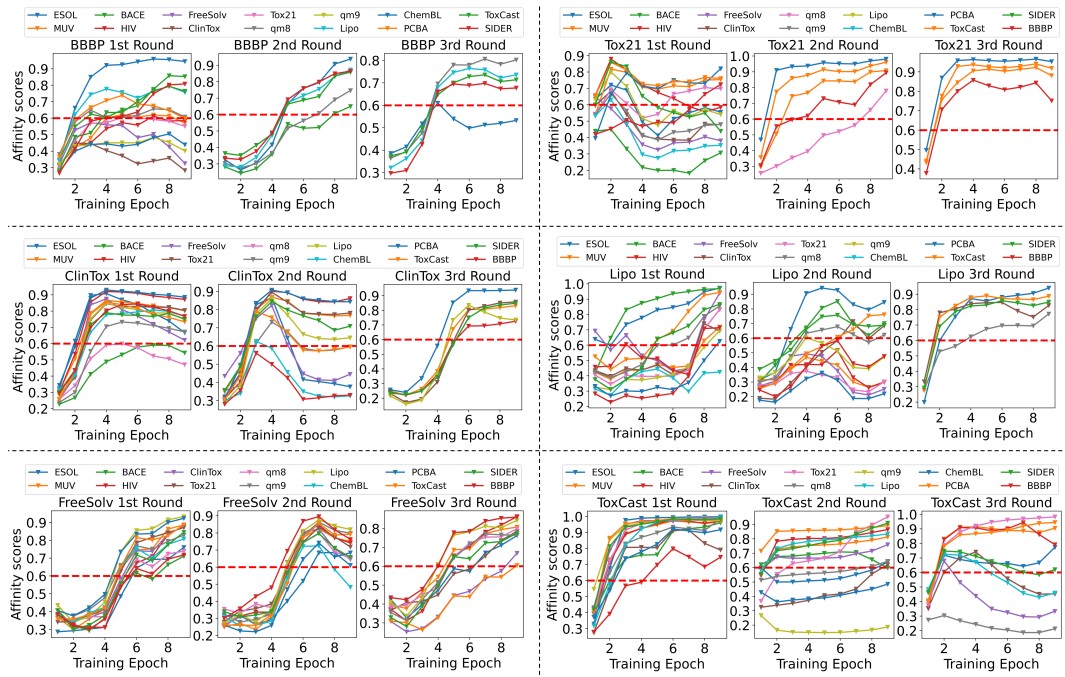

Figure 8: Learning curves of affinity scores where red dashed line represents threshold.

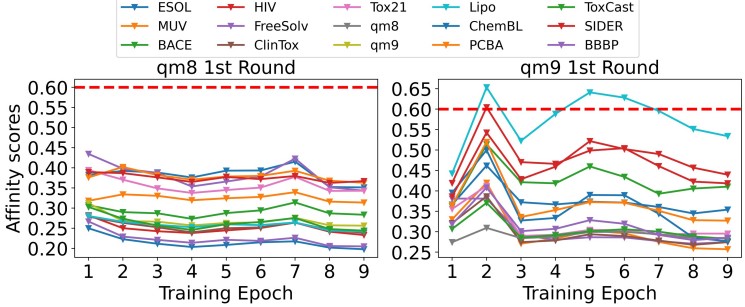

Figure 9: Learning curves for qm8 and qm9.

# E  Parameter Analysis

## E.1  Analysis on balance coefficient $\lambda$

To analyze the impact of the balance coefficient $\lambda$, we vary $\lambda$ in the range of $\{0.9, 0.7, 0.5, 0.3\}$ and pick BBBP, Tox21, and Lipo as examples. The learning curves of the affinity scores corresponding to different values of $\lambda$ are plotted in Fig.10. From the results, we observe that decreasing $\lambda$ led to lower affinity scores for the auxiliary datasets, which fails to effectively discriminate the affinity of individual auxiliary datasets. One reason for this is the instability introduced by parameter initialization and the per-step level computation of structure affinity scores. As a result, lower values of $\lambda$ cause the MolGroup to assign lower affinity scores to stabilize the training of the target dataset. In light of this, we suggest setting $\lambda$ to a high value, i.e., 0.9.

## E.2  Analysis on number of filtering rounds $R$

As shown in Fig.8, the auxiliary datasets with the highest affinity scores change as we progress through the different filtering rounds. To investigate the impact of the filtering rounds, we test the performance of the top-3 auxiliary datasets selected in each rounds. The comparison results are shown in Table 10 and the selected datasets are illustrated in Fig. 11. We observe that the auxiliary datasets selected in the 3rd round exhibit the best performance. This is attributed to the filtering

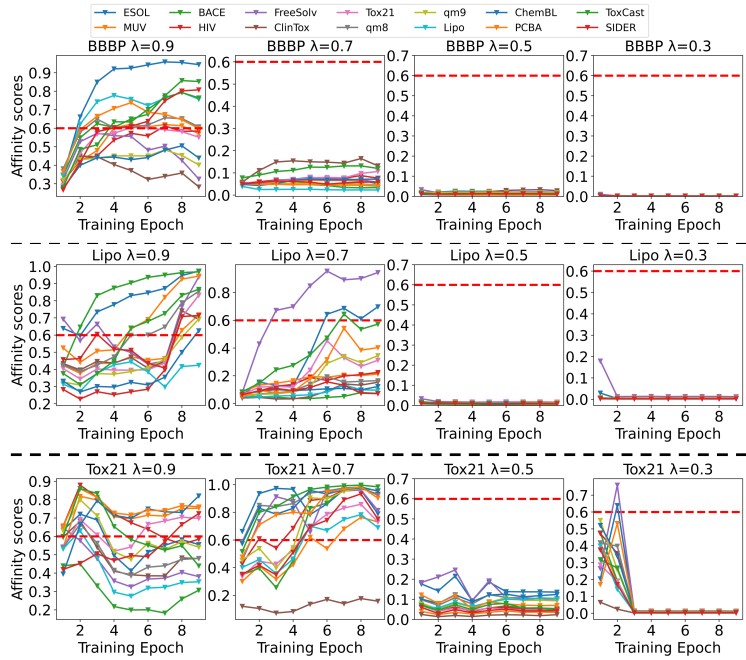

Figure 10: Learning curves of affinity scores with different $\lambda$.

process in the 1st and 2nd rounds, which removes negative datasets and helps alleviate interference. As a result, MolGroup can estimate the affinity of each auxiliary dataset more accurately, leading to improved performance on the target dataset.

| | BBBP | Tox21 | ClinTox | Lipo | FreeSolv | ToxCast |
|---|---|---|---|---|---|---|
| $R = 1$ | $67.94_{0.018}$ | $75.66_{0.004}$ | $57.66_{0.027}$ | $0.8013_{0.013}$ | $3.2288_{0.543}$ | $58.82_{0.007}$ |
| $R = 2$ | $67.99_{0.018}$ | $75.66_{0.004}$ | $57.66_{0.027}$ | $0.8020_{0.031}$ | $3.2280_{0.627}$ | $63.47_{0.003}$ |
| $R = 3$ | $\mathbf{68.36_{0.016}}$ | $\mathbf{75.66_{0.004}}$ | $\mathbf{59.77_{0.027}}$ | $\mathbf{0.7996_{0.007}}$ | $\mathbf{3.116_{0.279}}$ | $\mathbf{63.91_{0.005}}$ |

Table 10: Performance with different number of filtering rounds $R$.

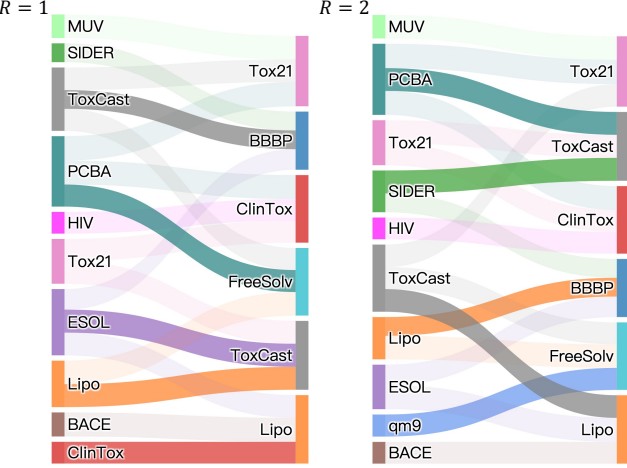

Figure 11: Auxiliary datasets with top-3 affinity scores with different $R$ where we highlight the different edges in these two rounds.

# F  Broader impact

**Impact on machine learning research.**    We propose a novel strategy that combines the routing mechanism with the meta gradient to quantify the impact of one dataset on another. Previously, the routing mechanism was used to increase the model capacity. Our proposed framework can inspire various extensive applications in machine learning, including neural architecture searching (NAS) and data-centric AI. Specifically, our framework enables the network to determine the optimal routing path through the meta gradient. This empowers the network to control and modify the layerwise architecture, leading to improved performance. Besides, it has the potential to enhance data-centric AI approaches by providing a tool to analyze and understand the relationship between different datasets or sub-datasets.

**Impact on biological research.**    We investigate the relationship between molecule datasets, and, focusing on both the structural and task dimensions. our findings shed light on how different molecule datasets impact each other. The analytical approach we employ can be extended to other biological data, such as protein and RNA. Furthermore, our method has the potential to be utilized for data instance filtering in the biological domain. This is particularly important given that biological data often contains various types of noise. By filtering out data instances with negative effects on downstream tasks, we can improve the quality and reliability of biological data used in research.

**Impact on the society.**    Our study has significant implications for society, particularly in the field of biomedicine and drug discovery. By understanding the relationship between molecule datasets and their impact on each other, we can gain insights into how different molecules interact and influence biological processes. It can aid in the development of more effective drugs and therapies by identifying molecules that have a positive impact on specific biological targets or diseases.

