# OpenReview forum: "Learning to Group Auxiliary Datasets for Molecule"
_NeurIPS.cc/2023/Conference — NeurIPS 2023 poster_

### Official Review · Reviewer_sjEd · 2023-07-03

**Soundness:** 3 good
**Presentation:** 3 good
**Contribution:** 3 good
**Rating:** 7
**Confidence:** 4

**Summary:**

This paper studies an interesting and meaningful problem, that is, how to make the best use of available molecule data for achieving superior performance on the prediction of molecule properties. This paper first conducts an extensive study on many widely used benchmark datasets and discovers some interesting patterns. Then the authors propose a routing-based method that considers both task similarity and molecule similarity to predict the best combination of auxiliary datasets for each target dataset.

**Strengths:**

1. The studied problem is interesting and meaningful because it is a very often case in the drug discovery pipeline.
2. The paper is clearly written and easy to follow.
3. The empirical study is well-designed and conducted, and the proposed method is well-motivated.

**Weaknesses:**

1. Computation cost will be high if there are many tasks involved.
2. Some datasets studied are essentially formed by multiple properties (e.g. Tox21, Clintox). It would be better if these properties are treated as one single task, instead of considered together.
3. This paper studies many datasets, and therefore it would be better if the authors can give a brief introduction to each of the datasets (e.g. what property is the dataset about?).

**Questions:**

1. What does $\mathbb{z}$ in Eqn. (1) mean? Is it the representation for a batch or the whole dataset?
2. Why the learnable paratmeter of $g_m(\cdot,\cdot)$ is not in Eqn. 7?
3. Is the proposed method sensitive to the threshold in Sec. 4.3 ?
4. In addition to task similarity and structural similarity, what are the other possible factors determining whether an auxiliary task is helpful or not? I hope the authors can discuss this.
5. What is the difference between search-based methods and group-based methods (line 237)?
6. As shown in Fig. 6, FreeSolv and ESOL are not quite helpful to each other. This seems strange --- FreeSolv is about Hydration Free Energy (HFE) and ESOL is about water solubility. Theoretically, these two properties should be strongly correlated as HFE is an important factor in water solubility. Do you have any comments on this result?

**Limitations:**

N.A.

---

> ### Author Rebuttal · Authors · 2023-08-10
>
> We sincerely thank you for your valuable comments on our paper. We will explain your concerns point by point.
>
> ```
> Q1: Computation cost will be high if there are many tasks involved.
> ```
>
> **Response**: It is indeed true that handling numerous datasets, especially when including large auxiliary sets, can consume significant computational resources. However, our proposed method can alleviate this by automatically grouping the datasets with the routing mechanism in a single model. This approach leads to improved efficiency compared with the other baseline models, as demonstrated by Fig.5(b).
>
> ```
> Q2: Some datasets studied are essentially formed by multiple properties (e.g. Tox21, Clintox). It would be better if these properties are treated as one single task, instead of considered together.
> ```
>
> **Response**: Thanks for your great suggestions! The current model only considers dataset-level information so it cannot fully exploit the sub dataset-level information in terms of task or structure. We will explore it in our future work.
>
> ```
> Q3: This paper studies many datasets, and therefore it would be better if the authors can give a brief introduction to each of the datasets (e.g. what property is the dataset about?).
> ```
>
> **Response**: Good suggestion! The introduction can be found in [MoleculeNet](https://moleculenet.org/). We will include it in the next version.
>
> ```
> Q4: What does 𝕫 in Eqn. (1) mean? Is it the representation for a batch or the whole dataset?
> ```
>
> **Response**: It represents the representations of the input batch. We apologize for any confusion and will make a revision in the next version.
>
> ```
> Q5: Why the learnable parameter of $g_m(\cdot,\cdot)$ is not in Eqn. 7?
> ```
>
> **Response**: Sorry for any confusion. $\alpha_m$ is produced by $g_m(\cdot,\cdot)$ and we only present the former in Eqn.7 for simplifying the expression. We will clarify the relationship between $\alpha_m$ and $g(\cdot, \cdot)$ in the revised version of the paper.
>
> ```
> Q6: Is the proposed method sensitive to the threshold in Sec. 4.3 ?
> ```
>
> **Response**: In our experiments, we found that setting a high threshold, e.g., 0.8, may lead to unstable selection results, as it potentially filters out most auxiliary datasets in the first or second round. A low threshold, like 0.2, requires more iterations for a more refined grouping. We have shown the learning curves of each iteration in Appendix D.3 with the threshold of 0.6, from which a stable selection process can be observed.
>
> ```
> Q7: In addition to task similarity and structural similarity, what are the other possible factors determining whether an auxiliary task is helpful or not? I hope the authors can discuss this.
> ```
>
> **Response**: Thanks for the suggestions! Besides the task and the structure, one significant factor is the strategy employed in training the model using multiple auxiliary datasets. Here we simply merge the datasets and sample them with equal possibility, which can potentially be extended. For example, selectively training different portions of the model parameters with specific datasets might lead to improved performance. We will thoroughly discuss this in the revised version of our manuscript.
>
> ```
> Q8: What is the difference between search-based methods and group-based methods (line 237)?
> ```
>
> **Response**: The distinction between search-based methods and grouping-based methods lies in two aspects:
>
> 1. Criterion: Grouping-based methods utilize a learnable metric to evaluate the affinity between datasets, such as the task embedding similarity of Task2vec. Conversely, search-based methods assess affinity by directly testing performance or by integrating fundamental features, such as similarities in fingerprint features.
> 2. Selection process: Grouping-based methods select the top-K auxiliary datasets based on the defined criterion. Search-based methods explore the auxiliary datasets through a breadth-first approach, retaining a candidate set at each level of exploration.
>
> ```
> Q9: As shown in Fig. 6, FreeSolv and ESOL are not quite helpful to each other. This seems strange --- FreeSolv is about Hydration Free Energy (HFE) and ESOL is about water solubility. Theoretically, these two properties should be strongly correlated as HFE is an important factor in water solubility. Do you have any comments on this result?
> ```
>
> **Response**: Although these two datasets share similar tasks, the difference in molecule structure distribution can still result in performance degradation, as demonstrated in Fig.7(a)(b). Besides, a similar task doesn't always lead to better performance, and actually, the structure distribution of ESOL dataset doesn’t exhibit a high correlation to the performance gain, as shown in Fig.3(b).

---

> > ### Comment · Reviewer_sjEd · 2023-08-18
> > **Response to the authors**
> >
> > Thank the authors for their reply. I have no more questions and would like to keep my rating unchanged.

---

> > > ### Author Response · Authors · 2023-08-20
> > > **Response to the reviewer sjEd**
> > >
> > > We appreciate your consideration and thoughtful feedback! The paper will be revised based on your suggestions and comments.

---

### Official Review · Reviewer_ohh4 · 2023-07-04

**Soundness:** 2 fair
**Presentation:** 3 good
**Contribution:** 3 good
**Rating:** 6
**Confidence:** 4

**Summary:**

The authors address the problem of, in a transfer learning, meta-learning, or few-shot learning setting involving molecules, which datasets might be most useful in providing positive auxiliary information that does not damage model performance on the task of interest.

The paper proposes a method, MolGroup, to identify the datasets which, if provided as auxiliaries to a model, most increase the score of a target dataset. The method makes use of a routing mechanism to select the optimal auxiliaries, and demonstrates improvement of both GIN and Graphormer models on some of 11 target datasets.

**Strengths:**


The method is novel and well-motivated, with routing mechanism and bilevel optimization which has not previously been applied in this setting.

The method does find auxiliary datasets which provide a relative improvement on the target datasets, where the gains are small but nonetheless present across the board. In addition the method is not computationally infeasible, and the mention of efficiency of the method as measured by wall clock time is valuable.

The experiments are comprehensive and carefully performed, taking into account SOTA modelling techniques when comparing final performance.

**Weaknesses:**

It is rather unclear initially whether the authors propose to calculate an affinity score based upon calculated task embedding and fingerprint distribution differences between datasets, or learn an affinity score as the gating score. Line 65-68 in particular are unclear on this point. Clarification would be very useful at this point in the manuscript.

The assertion that fingerprint and task embedding similarity are strictly different measures. In practice, task embeddings are highly dependent on the distribution of input features, regardless of labels, and therefore are rather similar to fingerprint embedding similarities. An earlier justification for this assertion in the manuscript would be very helpful.

While Figure 3 demonstrates that these correlation measures between auxiliary and target dataset are themselves correlated with relative improvement of performance, it is not especially convincing that the two correlation measures are distinct. For instance, some tasks show positive correlation for one measure and negative for another, others do not. It is not clear how to interpret this information. Expansion around these plots would be valuable.

It would be useful to see whether the method works on another domain outside of molecular property prediction -- have the authors considered this?

**Questions:**

A number of questions are raised in the "weaknesses" section above.

**Limitations:**

The authors do not discuss limitations.

---

> ### Author Rebuttal · Authors · 2023-08-10
>
> Thanks for your feedback on our work. We will address your main concern point by point.
>
> ```
> Q1: It is rather unclear initially whether the authors propose to calculate an affinity score based upon calculated task embedding and fingerprint distribution differences between datasets, or learn an affinity score as the gating score. Line 65-68 in particular are unclear on this point. Clarification would be very useful at this point in the manuscript.
> ```
>
> **Response**: We apologize for any confusion and will make a revision in the next version. In our work, we propose to learn the gate scores with a routing mechanism to quantify the affinity between datasets. Thank you for bringing this to our attention.
>
> ```
> Q2: The assertion that fingerprint and task embedding similarity are strictly different measures. In practice, task embeddings are highly dependent on the distribution of input features, regardless of labels, and therefore are rather similar to fingerprint embedding similarities. An earlier justification for this assertion in the manuscript would be very helpful.
> ```
>
> **Response**: Thank you for the suggestion! Actually, the task embedding is obtained using the GIN which applies extracted node and edge features rather than directly utilizing fingerprint features. It can inherently minimize the correlation between fingerprint and task embedding similarity, a fact evidenced by the substantial difference in their Pearson coefficients (0.16 vs 0.06). We will include this discussion in the next version to prevent any confusion.
>
> ```
> Q3: While Figure 3 demonstrates that these correlation measures between auxiliary and target dataset are themselves correlated with relative improvement of performance, it is not especially convincing that the two correlation measures are distinct. For instance, some tasks show positive correlation for one measure and negative for another, others do not. It is not clear how to interpret this information. Expansion around these plots would be valuable.
> ```
>
> **Response**: Thanks for your suggestions! Here are some examples using asymmetric KL divergence as our similarity measurement:
>
> 1. For the datasets Esol and Freesolv, the fingerprint similarity is 0.0212, while the task similarity is 0.4084. Their tasks are both related to water solubility but structural distributions vary substantially.
> 2. For the datasets Tox21 and SIDER, the fingerprint similarity is 0.8852, while the task similarity is 0.0062. The result reveals that the two datasets share similar structure distribution, yet their tasks differ significantly.
>
> We will incorporate more discussion in the revised version. Besides, we have included some case studies in Section 5.3, which shows the learned structure affinity scores among the datasets with the toxicity-related task (Tox21, ToxCast, and ClinTox).
>
> ```
> Q4: It would be useful to see whether the method works on another domain outside of molecular property prediction -- have the authors considered this?
> ```
>
> **Response**: Thanks for your suggestion! We believe that MolGroup can be potentially extended into other biomedical domains, such as protein and single-cell data, where data distribution varies and label annotations are costly. We will keep exploring it in our future work.
>
> ```
> Limitations: The authors do not discuss limitations.
> ```
>
> **Response**: The limitation is included in the Conclusion section. We apologize for any oversight and will ensure greater clarity in the revised version.

---

### Official Review · Reviewer_NGKd · 2023-07-08

**Soundness:** 3 good
**Presentation:** 3 good
**Contribution:** 3 good
**Rating:** 6
**Confidence:** 3

**Summary:**

The paper addresses the challenge of limited annotations in small molecule datasets and proposes a method called MolGroup to identify auxiliary datasets that can benefit the target dataset when jointly trained. MolGroup utilizes a routing mechanism optimized through a bi-level optimization framework to separate dataset affinity into task and structure affinity. The proposed method demonstrates its effectiveness in predicting the optimal combination of auxiliary datasets for each target dataset and outperforms existing baselines.

**Strengths:**

1. The paper provides a clear motivation for the problem of limited annotations in small molecule datasets and the challenges associated with incorporating auxiliary datasets. This highlights the practical relevance of the research.
2. The proposed MolGroup method is well-explained and builds on the insights obtained through empirical analysis. Particularly, the preliminary study on the relative improvement and the similarities is interesting.
3. The extensive experiments demonstrate the efficiency and effectiveness of MolGroup.

**Weaknesses:**

1. The Pearson coefficients presented in Figure 3 are relatively low, all below 0.5. This raises doubts about the claim that the combination of task and structure leads to better performance. There is a potential risk of negative transfer that could negatively impact the main task's performance. It would be better to provide further insights or explanations to address this concern.
2. Given that the authors propose to use meta learning to strengthen the learning process, it is suggested that an ablation study be conducted to demonstrate the effectiveness of this approach. Comparing the performance with and without meta learning would provide a clearer understanding of its contribution to the proposed MolGroup method.
3. In line 187, the authors propose to assign learnable embeddings for the tasks. Why not use Task2vec to generate the task embedding, as is done in Section 3? The reasoning behind this choice is not adequately explained. It would be beneficial for the authors to provide a justification for this decision and discuss any potential implications or advantages of using learnable embeddings over Task2vec.

**Questions:**

Please see the weaknesses

**Limitations:**

Yes

---

> ### Author Rebuttal · Authors · 2023-08-10
>
> Thanks for your feedback on our work! We will explain your concerns point by point.
>
> ```
> Q1: The Pearson coefficients presented in Figure 3 are relatively low, all below 0.5. This raises doubts about the claim that the combination of task and structure leads to better performance. There is a potential risk of negative transfer that could negatively impact the main task's performance. It would be better to provide further insights or explanations to address this concern.
> ```
>
> **Response**: The single metric(fingerprint/task similarity) indeed doesn’t correlate strongly with the performance gain. However, **the primary purpose** of this analysis is to demonstrate that integrating both structural and task information yields a stronger correlation with performance gains rather than proposing a novel metric to predict affinity.
>
> Besides, compared with these single metrics, the proposed MolGroup can better capture the affinity between datasets in terms of task and structure, as demonstrated in Table1,2 and Fig.7(a). Combining both sides of information can lead to a more comprehensive understanding of the affinity between datasets. We will add more analysis in the revised version.
>
> ```
> Q2: Given that the authors propose to use meta learning to strengthen the learning process, it is suggested that an ablation study be conducted to demonstrate the effectiveness of this approach. Comparing the performance with and without meta learning would provide a clearer understanding of its contribution to the proposed MolGroup method.
> ```
>
> **Response**: Thanks for your suggestion! We have presented the learning curves of the affinity scores without utilizing the bi-level framework in Fig.5(a). The result shows a homogeneous distribution among the auxiliary datasets, indicating that the model struggles to distinguish the affinity of the auxiliary dataset without bi-level training. We will conduct more analysis to provide a more comprehensive understanding in future work.
>
> ```
> Q3: In line 187, the authors propose to assign learnable embeddings for the tasks. Why not use Task2vec to generate the task embedding, as is done in Section 3? The reasoning behind this choice is not adequately explained. It would be beneficial for the authors to provide a justification for this decision and discuss any potential implications or advantages of using learnable embeddings over Task2vec.
> ```
>
> **Response**: As demonstrated in Fig. 3(a) and Tables 1 and 2, Task2vec exhibits a weak correlation with the performance gain (0.05 Pearson coefficient). Additionally, the performance of the model trained with the auxiliary datasets selected by Task2vec is poor. We will add more discussions in the revised version.

---

> > ### Comment · Reviewer_NGKd · 2023-08-19
> > **Thanks for the rebuttal**
> >
> > My concerns are addressed and I would like to increase my score to 6.

---

> > > ### Author Response · Authors · 2023-08-20
> > > **Response to the reviewer NGKd**
> > >
> > > We greatly appreciate your insightful comments and suggestions, which have significantly improved our paper.

---

### Official Review · Reviewer_VvkN · 2023-07-08

**Soundness:** 2 fair
**Presentation:** 3 good
**Contribution:** 2 fair
**Rating:** 4
**Confidence:** 4

**Summary:**

The paper proposed MolGroup, a dataset grouping method designed to aid molecule property prediction. Motivated by preliminary empirical analysis, MolGroup separates the dataset affinity into task and structure affinity, and uses a routing mechanism to quantify the affinity between a pair of datasets. The routing mechanism is optimized through a bi-level optimization framework.  Experiments on 11 target molecule datasets show that MolGroup yields a ~4% increase across two architectures.

**Strengths:**

**Originality**

The separation of task and structure affinity is novel. The application of bi-level optimization for quantifying dataset affinity is novel.

**Clarity**

The paper is well-structured and easy to follow.

**Weaknesses:**

**Quality**

In Fig. 3a, I don't think we can draw the conclusion that structure and task affinities are compensatory. If we remove the outliers, the points seem randomly distributed, which means that the affinities and the relative improvements are not correlated. This phenomenon might significantly undermine the subsequent arguments.

**Significance**

I agree that molecular property prediction is a very important task with limited data. But if putting this much effort and data result in a mere 4% performance increase, how do you convince the research community that we shall continue on this direction?

**Questions:**

1. Could you elaborate/reiterate on how you train the final model? Why do you not use the routing mechanism in your final model (I suppose it is due to performance issues)?
2. Please address the quality and significance issue stated above.

**Limitations:**

I do not see any significant, unreported negative societal impact.

---

> ### Author Rebuttal · Authors · 2023-08-10
>
> We sincerely thank you for the insightful comments! We will address your concerns point by point.
>
> ```
> Q1: In Fig. 3a, I don't think we can draw the conclusion that structure and task affinities are compensatory. If we remove the outliers, the points seem randomly distributed, which means that the affinities and the relative improvements are not correlated. This phenomenon might significantly undermine the subsequent arguments.
> ```
>
> **Response**: Thanks for your suggestions! The outliers are the cases of dataset pair with significant improvements or degradations, such as the FreeSolv dataset with others. Here we exclude all the datasets exhibiting huge performance changes, i.e., FreeSolv, qm8, and qm9, and the regression curves are included in the attachment. It can be observed that the combination of structure and task information still exhibits a stronger correlation compared to the use of the individual one, which is consistent with our analysis shown in the paper.
>
> Besides, the advantage of applying both task and structure affinity can be further demonstrated by the case studies in Section 5.3, where the structure affinity score explains why datasets with similar tasks cannot benefit from each other. We will add more analysis in the revised version.
>
> ```
> Q2: I agree that molecular property prediction is a very important task with limited data. But if putting this much effort and data result in a mere 4% performance increase, how do you convince the research community that we shall continue on this direction?
> ```
>
> **Response**: Actually improving the small molecule datasets with a 4% gain is non-trivial and challenging, which can also be observed in the previous state-of-the-art methods[1,2]. These previous methods employ large parameter space and additional information (i.e., 3D structure and large pretrained dataset) to achieve a relative improvement of 4%-6%, comparable to ours.
>
> Moreover, our proposed method has the additional advantage of efficiency compared with search-based methods, as illustrated in Figure 5 (b). The grouping results are model-agnostic and hold for various backbone models, as demonstrated by Tables 1 and 2. We would like to highlight that the model trained with the grouping dataset can still benefit from other techniques like pre-training.
>
> [1] Zhou, G., Gao, Z., Ding, Q., Zheng, H., Xu, H., Wei, Z., ... & Ke, G. (2023). Uni-Mol: a universal 3D molecular representation learning framework.
> [2] Rong, Y., Bian, Y., Xu, T., Xie, W., Wei, Y., Huang, W., & Huang, J. (2020). Self-supervised graph transformer on large-scale molecular data. *Advances in Neural Information Processing Systems*, *33*, 12559-12571.
>
> ```
> Q3: Could you elaborate/reiterate on how you train the final model? Why do you not use the routing mechanism in your final model (I suppose it is due to performance issues)?
> ```
>
> **Response**: We first merge all the selected auxiliary datasets together and train an initial model on this merged dataset. Note that the initial model, i.e., the final model, doesn’t employ the routing mechanism which is only utilized for auxiliary grouping datasets. The data is sampled from each dataset with equal probability during the training.
>
> Our proposed routing mechanism indeed can be used to train the final model on all the datasets and adjust the influence between datasets in an end-to-end manner. But such a method suffers from two limitations:
>
> 1. Training a model with more datasets will consume significantly more computational resources and time.
> 2. Our empirical study shows that the model trained without the negative auxiliary datasets outperforms significantly the model trained with negative datasets, although when it is equipped with a routing mechanism. It can be demonstrated in the following table:
>
> | GIN | BBBP(↑) | toxcast(↑) | tox21(↑) | esol(↓) | freesolv(↓) |
> | --- | --- | --- | --- | --- | --- |
> | Only-target | 0.6662(±0.0284) | 0.6069(±0.0102) | 0.7423(±0.0057) | 1.5635(±0.0408) | 3.8421(±1.5796) |
> | Molgroup | 0.6836(±0.0163) | 0.6391(±0.0058) | 0.7566(±0.0044) | 1.4028(±0.0372) | 3.1166(±0.2790) |
> | Final Model with routing mechanism | 0.6683(±0.0307) | 0.5706(±0.0114) | 0.6937(±0.0180) | 3.0400(±0.1383) | 5.4833(±1.3547) |
>
> MolGroup can be considered a **hard version** of the final model including the routing mechanism. Rather than incrementally eliminating negative training signals, it directly filters out negative datasets. Furthermore, a final model trained with the routing mechanism fails to yield grouping results that can be transferred to other models.

---

> > ### Comment · Reviewer_VvkN · 2023-08-14
> > **Reply to the authors**
> >
> > Thank you for your clarifications.
> >
> > Overall, I think your paper is solid but contains too many arbitrary choices to be considered elegant.
> > E.g., the method is based on the usefulness of structure and task affinities.
> > The former metric is based on molecular fingerprints, which could be viewed as an additional data modality.
> > It is possible that combining GIN/Graphphormer with fingerprints could yield simpler models with comparable improvement (this is purely a hypothesis, of course).
> > Therefore, I maintain my initial evaluation.

---

> > > ### Author Response · Authors · 2023-08-17
> > > **Response to the reviewer**
> > >
> > > Thanks for your response! We’d like to clarify that the fingerprint feature is **only** used for grouping datasets, and is not incorporated during the final model training. Tables 1 and 2 in our paper show that adding FP features to beam search cannot consistently improve results compared to the original one. This suggests that the FP features do not guarantee better performance.
> > >
> > > Here we compare the performance of MolGroup with the model trained with FP features, which is shown in the following table. It can be found that the FP features fail to consistently improve the performance of all the datasets (Only-target vs Only-target+FP). Besides, our proposed MolGroup can outperform or matches the performance of the model trained with FP features in most cases.
> > >
> > > | GIN | BBBP(↑) | toxcast(↑) | tox21(↑) | esol(↓) | freesolv(↓) |
> > > | --- | --- | --- | --- | --- | --- |
> > > | Only-target | 0.6662(±0.0284) | 0.6069(±0.0102) | 0.7423(±0.0057) | 1.5635(±0.0408) | 3.8421(±1.5796) |
> > > | Only-target+FP | 0.6624(±0.0169) | 0.6121(±0.0081) | 0.7278(±0.0076) | 1.2309(±0.1014) | 2.2646(±0.2444) |
> > > | Molgroup | 0.6836(±0.0163) | 0.6391(±0.0058) | 0.7566(±0.0044) | 1.4028(±0.0372) | 3.1166(±0.2790) |

---

> > > > ### Comment · Reviewer_VvkN · 2023-08-20
> > > > **Response to the authors**
> > > >
> > > > Thank you to your timely response. I really appreciate your efforts. Unfortunately the story of your paper still could not fully convince me. I will keep my score, but wish you good luck in getting your paper accepted.

---

> > > > > ### Author Response · Authors · 2023-08-20
> > > > > **Reponse to the reviewer VvkN**
> > > > >
> > > > > Thanks for your time to review and provide feedback on our paper. Regardless of the outcome, we truly appreciate your insights and understand your perspective. Thank you once again.

---

### Official Review · Reviewer_B1V2 · 2023-07-09

**Soundness:** 3 good
**Presentation:** 2 fair
**Contribution:** 2 fair
**Rating:** 6
**Confidence:** 3

**Summary:**

This paper investigates how different molecule datasets affect each other’s learning, considering both task and structure aspects. It proposes a routing-based molecule grouping method to calculate the affinity scores of each auxiliary dataset based on the graph structure and task information, and select the auxiliary datasets with high affinity. The selected datasets are then  combined and fed into the downstream model. Experiments show a large improvement for GIN/Graphormer trained with the selected group of molecule datasets.

**Strengths:**

1. This paper studies how to select additional auxiliary molecule datasets to improve the prediction performance on the target dataset. Annotated molecule dataset is difficult to obtain and naively introduce auxiliary molecule datasets may result in negative transfer. This paper investigates an interesting problem how to properly select auxiliary molecule datasets. The authors design a dataset grouping method for molecules, which considers both task and structure aspects.

2. Design a routing mechanism to quantify the affinity between two datasets and bi-level optimization framework used to update the routing mechanism through meta gradient. The proposed routing function can comprehensively measure the affinity from two perspectives: task and graph structure. The bi-level optimization uses the parameters updated by the auxiliary dataset as a signal to guide the learning of the routing mechanism.

3. Experimental results how strong improvement of the baseline methods. The proposed method outperforms all the baseline methods and consistently improves the performance of the backbone model, with an average relative improvement of 6.47% across all datasets. Extra analysis of the proposed method is presented.

**Weaknesses:**

1. The authors did not study how auxiliary molecule datasets impact different models. If the proposed method is model-agnostic?

**Questions:**

If the proposed method is model-agnostic?

**Limitations:**

Yes.

---

> ### Author Rebuttal · Authors · 2023-08-10
>
> Thanks for your valuable comments on our paper! Here we additionally present the performance of GCN (#layer=2, hidden dim=300, dropout rate=0.5) with different grouping methods in the following table, where a consistent improvement can be observed. Such a phenomenon also aligns with the performance of GIN and Graphormer shown in Tables 1 and 2 in our paper, demonstrating that the grouping can be transferred to different models. We appreciate your suggestion and will conduct a more comprehensive evaluation in our future work.
>
> | GCN    | BBBP(↑) | ToxCast(↑) | Tox21(↑) | ESOL(↓)  | FreeSolv (↓)|
> | -------- | ------- |  ------- | ------- | ------- | ------- |
> | Only-target   |   0.6309(±0.0136)|	0.6253(±0.0086)|	0.7427(±0.0061)|	1.4720(±0.0277)|	3.3941(±0.2063)|
> | TAG   |   0.6159(±0.0123)|	0.6164(±0.0037)|	0.6996(±0.0058)|	1.5028(±0.0423)|	2.7482(±0.2939)|
> | MolGroup  |   0.6369(±0.0084)|	0.6290(±0.0062)|	0.7527(±0.0054)|	1.3745(±0.0297)|	2.5496(±0.2700)|

---

### Author Rebuttal · Authors · 2023-08-10

We thank the reviewers for noting that we propose a novel method (VvkN,ohh4) to address a meaningful problem (B1V2,VvkN,sjEd) with a clear motivation (VvkN,NGKd,ohh4,sjEd), and the paper is well-written and easy to follow (VvkN,NGKd,ohh4,sjEd). We further summarize our key contributions as follows:

1. We study a new angle to improve the performance of the molecule dataset with limited annotations by utilizing the auxiliary datasets with high affinity. The strategy is compatible with the other training strategy such as pertaining.
2. We conduct a study to analyze how different molecule datasets affect each other in terms of task and structure similarity. This investigation has led to some interesting findings, such as the compensatory relationship between structure and task, and the performance gains achieved by integrating both aspects.
3. A routing-based molecule grouping method optimized by a bi-level learning framework is proposed which achieves the SOTA performance across 11 molecule datasets.

Besides, we include the regression curves between relative improvement and the task/structure similarity measurement excluding the datasets with huge performance changes in the attachment (reviewer VvkN).

---

### Decision · Program_Chairs · 2023-09-21

**Decision:**

Accept (poster)

**Comment:**

The paper addresses the limited data problem in molecule property prediction by proposing a novel method, MolGroup, which effectively leverages auxiliary datasets to improve performance on target datasets. The proposed method includes a routing mechanism and bi-level optimization, which is well-motivated and clearly presented. Extensive experiments are also comprehensive and carefully conducted, demonstrating the effectiveness of the proposed method. While there are some concerns raised by the reviewers, such as the interpretation of correlation and the ablation study, the overall contributions and significance of the paper outweigh these concerns. The authors are encouraged to revise the paper with respect to the comments in the final version.